# Application of the Bland–Altman and Receiver Operating Characteristic (ROC) Approaches to Study Isotope Effects in Gas Chromatography–Mass Spectrometry Analysis of Human Plasma, Serum and Urine Samples

**DOI:** 10.3390/molecules29020365

**Published:** 2024-01-11

**Authors:** Dimitrios Tsikas

**Affiliations:** Core Unit Proteomics, Institute of Toxicology, Hannover Medical School, 30623 Hannover, Germany; tsikas.dimitros@mh-hannover.de

**Keywords:** Bland–Altman, chromatography, isotope effects, GC-MS, retention time, ROC

## Abstract

The Bland–Altman approach is one of the most widely used mathematical approaches for method comparison and analytical agreement. This work describes, for the first time, the application of Bland–Altman to study ^14^N/^15^N and ^1^H/^2^H (D) chromatographic isotope effects of endogenous analytes of the L-arginine/nitric oxide pathway in human plasma, serum and urine samples in GC-MS. The investigated analytes included arginine, asymmetric dimethylarginine, dimethylamine, nitrite, nitrate and creatinine. There was a close correlation between the percentage difference of the retention times of the isotopologs of the Bland–Altman approach and the area under the curve (AUC) values of the receiver operating characteristic (ROC) approach (*r* = 0.8619, *p* = 0.0047). The results of the study suggest that the chromatographic isotope effects in GC-MS result from differences in the interaction strengths of H/D isotopes in the derivatives with the hydrophobic stationary phase of the GC column. D atoms attenuate the interaction of the skeleton of the molecules with the lipophilic GC stationary phase. Differences in isotope effects in plasma or serum and urine in GC-MS are suggested to be due to a kind of matrix effect, and this remains to be investigated in forthcoming studies using Bland–Altman and ROC approaches.

## 1. Introduction

In gas chromatography (GC) and reversed-phase liquid chromatography (LC), isotopologs differ in their retention times, with deuterated compounds having, as a rule, smaller retention times (*t*_R_) than their protiated analogs [1]. The slightly reduced molecular volume of ^2^H-labeled compounds compared to their non-labeled analogs is assumed to be responsible for this phenomenon [2]. The *t*_R_ of the isotopologs is the main parameter for quantitate H/D isotope effects [3]. One method to calculate the H/D isotope effect (IE) is to divide the *t*_R_ of the protiated analyte *t*_R(H)_ by the *t*_R_ of the deuterated analyte *t*_R(D)_ (Formula (1)). The difference in the retention times can be used to estimate the extent of the isotope effect δ_(H/D)_ (Formula (2)). The closer the IE value to the unity (1.0000), the lower/weaker the isotope effect between the isotopologs. The smaller the difference δ_(H/D)_ in the retention times *t*_R(H)_ and *t*_R(D)_, the higher/stronger the isotope effect between the isotopologs. These issues are also valid for isotopes of other elements, including ^14^N and ^15^N.
IE = *t*_R(H)_/*t*_R(D)_(1)
δ_(H/D)_ = *t*_R(H)_ − *t*_R(D)_(2)
µ_(H/D)_ = 1/2 × (*t*_R(H)_ + *t*_R(D)_)(3)
δ (%) = 2 × ([*t*_R(H)_ − *t*_R(D)_]/[*t*_R(H)_ + *t*_R(D)_]) × 100 (4)

The Bland–Altman approach is useful for the comparison of methods with comparable analytical performance [4,5]. It is a graphical approach which examines the relationship between the difference (δ) of the values obtained by two methods and the average (μ) of the methods. In a variant of the Bland–Altman method, the percentage difference δ (%) of the two methods is plotted versus the average of these methods (Formulas (3) and (4)). One may expect that the Bland–Altman approach would be useful in calculating the absolute and percentage difference of the retention times of isotopologs as measures of isotope effects. The Bland–Altman method has been sporadically used in this area, including δ(^18^O) isotope ratios [6,7,8]. The receiver operating characteristic (ROC) approach is another graphical plot that is widely used in several disciplines, notably including clinical chemistry [9,10]. The utility of the ROC approach in comparison to methods in analytical chemistry has been demonstrated [5]. The ROC approach is useful for evaluating the agreement/disagreement between the isotopologs. 

In the present work, the Bland–Altman and ROC approaches were used for the first time to investigate the H/D and ^14^N/^15^N isotope effects of endogenous substances in human plasma, serum and urine samples. The analytes considered in the study belong to the L-arginine/nitric oxide pathway [11]. They include nitrite and nitrate, the major metabolites of nitric oxide (NO), L-arginine (Arg) and asymmetric dimethylarginine (ADMA), the endogenous substrate and inhibitor of NO synthase, respectively [11], and dimethylamine (DMA), the major urinary metabolite of ADMA [12]. In addition, creatinine serves as an analyte that is commonly used to correct for the excretion of endogenous substances in urine collected by spontaneous micturition. The above-mentioned analytes were measured by gas chromatography–mass spectrometry (GC-MS) after proper chemical derivatization. The chemical structures of the derivatives of the protiated (unlabeled) and ^2^H- or ^15^N-labelled analytes are shown in Figure 1. 

## 2. Methods

### 2.1. GC-MS Analyses in Human Plasma, Serum and Urine Samples

The human plasma, serum and urine samples analyzed in the present work were collected in previous clinical studies of the author’s group and cooperating groups after approval by the local ethics committees [13,14,15,16,17,18,19]. The studies were conducted in line with the ethical principles of the Declaration of Helsinki [20]. The COVID-19 study [17] was approved by the local ethics committees (9948_BO_K_2021 Hannover Medical School; 29/3/21 University Medical Center Göttingen). The ASOS study was approved by the Health Research Ethics Committee of North-West University (NWU-00007-15-A1) [18]. 

Nitrite and nitrate were analyzed simultaneously by GC-MS as described previously, using commercially available ^15^N-nitrite and ^15^N-nitrate as internal standards, respectively [13]. Creatinine was analyzed via GC-MS as described elsewhere using commercially available [*methylo*-^2^H_3_] creatinine as internal standard [14]. Arg and ADMA were analyzed by GC-MS, using in situ prepared trideuteromethyl ester as described previously [15]. DMA was analyzed by GC-MS using [*dimethylo*-^2^H_3_]DMA (d_6_-DMA) as the internal standard [16]. Metformin (METF) was analyzed by GC-MS using [*dimethylo*-^2^H_3_]metformin (d_6_-METF) as the internal standard [19]. The isotopic purity in the stable-isotope-labeled analogs was at least 99% at ^2^H and ^15^N. The concentrations of the internal standards were 1 µM for ADMA in plasma and serum; 20 µM in urine; 50 µM for Arg in plasma and serum; 10 mM and 1 mM for creatinine in urine and serum, respectively; 1 mM for DMA in urine; 4 µM for nitrite in plasma and serum and 8 µM in urine; 40 µM for nitrate in plasma and 800 µM in urine.

GC-MS analyses were performed on the apparatus model ISQ from ThermoFisher (Dreieich, Germany), which was equipped with a fused silica capillary OPTIMA-17 column (15 m × 0.25 mm, 0.25 µm film thickness) from Macherey-Nagel (Düren, Germany). Helium and methane were, respectively, used as carrier and reagent gases for negative-ion chemical ionization. Analyses were performed by selected-ion monitoring (SIM) for nitrite, nitrate, creatinine, DMA, ADMA and Arg. Aliquots of 1 µL toluene (for amino acids, DMA and metformin) or ethyl acetate (for nitrite, nitrate and creatinine) extracts of the derivatives were injected in the splitless mode. Ions were detected after conversion to electrons by using an electron multiplier. Different oven temperature programs were used, starting either at 40 °C or 70 °C (for nitrite, nitrate and creatinine).

### 2.2. Calculations

Isotope effect values were calculated using Formula (1). The difference (in min) between the retention times was calculated by Formula (2). The difference in the retention times was multiplied by 60 to obtain the outcome in seconds. In the Bland–Altman approach, the percentage difference (δ (%)) was plotted versus the average (µ_(H/D)_). The term bias (%) in the regular Bland–Altman approach corresponds to δ (%). The receiver operating characteristic (ROC) approach was used to determine the area under the curve (AUC) values by using the retention times of the isotopologs. The ratios of the peak areas (PAR) of endogenous analytes and the respective internal standards were calculated and used to test for potential correlations between the difference δ and the PAR values. The Wilcoxon matched-pairs signed-rank test was used to test statistical differences in the retention times of isotopologs.

### 2.3. Statistical Analyses and Data Presentation

GraphPad Prism Version 7 for Windows (GraphPad Software, San Diego, CA, USA) was used for statistical analyses and preparation of graphs, including the Bland–Altman and ROC plots. The ROC approach was used to calculate AUC values and evaluate agreement/disagreement between the isotopologs. AUC values are reported as mean with standard error. The Wilcoxon matched-pairs signed-rank test was used in two-tailed paired analyses. A *p*-value of <0.05 was considered significant. Chemical structures of the investigated derivatives of the isotopologs were drawn using ChemDraw 15.0 Professional (PerkinElmer, Germany).

## 3. Results

### 3.1. Bland–Altman and ROC Approaches to Study Isotope Effects in GC-MS in Biological Samples 

The primary results of the present study are listed in Table 1. The secondary results obtained from the application of the Bland–Altman and ROC approaches are summarized in Table 2 and shown in Figure 1. 

The retention times of all derivatives were measured with high precision (Table 1). The ^2^H and ^15^N isotopologs had smaller retention times than their ^1^H and ^14^N counterparts. Yet, the differences in the retention times were larger for the ^2^H analogs. The IE values for the ^15^N derivatives of nitrite and nitrate in plasma and urine were practically 1.0000, and the difference in the retention times was not higher than 0.18 s. The highest IE and δ values were observed for the PFBz derivative of d_6_-DMA, i.e., 1.007 and 1.5 s, respectively. The concentrations of the stable-isotope labeled analogs, which were added to the plasma and urine samples, were all relevant for the respective biological samples. This is indicated by the measured PAR values for all analytes, which ranged between 0.2 and 2.4. Weak correlations after Spearman between the PAR and δ values were observed for some analytes, indicating a very weak dependence of δ upon the endogenous analyte concentrations in the plasma and urine samples analyzed. 

The application of the Bland–Altman approach resulted in δ (%) values ranging between 0.714% for DMA in urine and 0.00841% for the PFB derivative of nitrite in plasma (Table 2). The application of the ROC approach resulted in AUC values ranging between 1.0000 for creatinine in urine and ADMA in plasma and 0.5414 for nitrite in plasma. The AUC values for ADMA and nitrate were higher in the plasma compared to urine samples, whereas the AUC value for nitrite was lower in plasma compared to urine. In urine, the δ (%) values (*y*) increased linearly with the average retention time (*x*) of the ^14^N/^15^N isotopologs (*y* = −200 + 44 × *x*, *r*^2^ = 1.000, *p* < 0.0001), indicating positive proportional error [5]. 

We tested for potential correlation between the AUC-ROC and Bland–Altman δ (%) values. We found a strong correlation after Spearman between these approaches: *r* = 0.862, *p* = 0.005 (Figure 1A). This Figure also illustrates the AUC-ROC and δ (%) differences for ADMA, nitrate and nitrite in the plasma and urine samples. Omitting the DMA values, a linear regression analysis between the AUC (*y*) and δ (%) (*x*) values resulted in a straight line with the regression equation *y* = 0.59 + 2.38 × *x*, *r*^2^ = 0.8673 (Figure 1B). 

In a further, recently performed study [17], we analyzed paired serum and urine samples of 85 volunteers who formerly had COVID-19 or were living with long-COVID-19. Nitrate, nitrite and creatinine were simultaneously analyzed by GC-MS on the apparatus ISQ and a 15 m long GC OPTIMA-17 column, as described previously [18]. The concentrations of the internal standards were 10 mM for creatinine in urine and 100 µM in serum, 4 µM for nitrite in serum and 8 µM in urine as well as 40 µM for nitrate in serum and 800 µM in urine. Instead of toluene [13,14], ethyl acetate [21] was used for the extraction of the PFB derivatives, and 1 µL aliquots of ethyl acetate extracts were injected in the splitless mode. The results of these analyses are summarized in Table 3.

The retention times of the corresponding isotopologs differed statistically significantly from each other. The values of IE, δ and AUC differed for the isotopologs in urine and serum, as well as when compared to urine with serum. The highest IE, δ and AUC values were observed for creatinine and the lowest were observed for nitrite in serum (Table 3). Figure 2 shows the relationship between the AUC-ROC and Bland–Altman values with respect to the retention times of the isotopologs in the serum and urine samples. 

### 3.2. Isotope Effects as a Measure of Matrix Effects in GC-MS: Proof-of-Concept Studies

Matrix effects are very common in LC-MS/MS, and methods have been proposed, with their measurements implemented in bioanalysis [22,23,24,25,26]. Matrix effects have been sporadically reported in GC-based methods, including GC-MS and GC-MS/MS [27,28,29,30]. Matrix-induced ion suppression effects occur both in electron ionization (EI) and NICI, yet the underlying mechanisms have not yet been explained thus far [26]. Stable-isotope-labeled analogs have been used in GC [31] and LC-MS/MS [32] to minimize matrix effects. To the best of our knowledge, isotopologs have not been used to quantify matrix effects in GC-MS or GC-MS/MS [26]. Given the observations of different IE and δ values for some analytes in serum and urine samples in the present study, we tested the utility of isotopologs to quantify matrix effects in GC-MS.

#### 3.2.1. GC-MS Analysis of Dimethyl Amine

We analyzed, via GC-MS, d_0_-DMA and d_6_-DMA after extractive derivatization with pentafluorobenzoyl chloride/toluene (Figure 1) [16]. Human urine samples (U, *n* = 80), a 67 mM potassium phosphate buffer of pH 7.0 (B, *n* = 33), a 20 mM Na_2_CO_3_ solution (C, *n* = 33) and deionized water (W, *n* = 33) were treated as follows:(1)A total of 10 µL of a 1 mM d_6_-DMA solution was introduced into autosampler glass vials;(2)A total of 10 µL of U, B, C or W was added (B, C and W contained d_0_-DMA at 0, 100 and 500 µM);(3)A total of 90 µL of W was added;(4)A total of 100 µL of 20 mM Na_2_CO_3_ was added.

d_6_-DMA was used as the internal standard at a fixed final concentration of 1000 µM in all matrices. The d_0_-DMA concentrations in B, C and W were 0, 100 and 500 µM. After derivatization, 1 µL aliquots of toluene extracts were injected splitless, and SIM of *m*/*z* 240 for d_0_-DMA and *m*/*z* 246 for d_6_-DMA was performed (ISQ apparatus, 15 m long OPTIMA-17 column). The results of this experiment are summarized in Table 4.

The PA of *m*/*z* 246 for d_6_-DMA-PFBz varied by 7.3%. In the urine samples (*n* = 80), the PAR of *m*/*z* 240 for d_0_-DMA to *m*/*z* 246 for d_6_-DMA ranged between 0.1 and 1.6 (mean, 0.608 ± 0.26).

The retention times of the isotopologs differed in all matrices but did not result in different IE values. This parameter was not further investigated. Statistically significant differences with respect to δ were found between urine (U) and buffer (B), as well as between urine (U) and water (W) by the Mann–Whitney test and the ROC approach. The highest δ value was observed for the carbonate solutions of DMA (C). The DMA solutions in B, C and W are more comparable among themselves than with the U samples, which were diluted 10-fold with water and carbonate. The experiment described above is a very simple simulation of potential matrix effects on isotope effects in GC-MS. A modification of this simulation, for instance, by using undiluted urine or urine diluted to varying degrees, would be more meaningful. Whether the Bland–Altman approach or the ROC approach is able to provide more definite results remains to be investigated. The Bland–Altman approach is expected to be more promising because of its higher versatility. 

#### 3.2.2. GC-MS Analysis of Metformin

Standard curves were prepared for d_0_-metformin (d_0_-METF) in human urine (U) and serum (S) samples in relevant metformin concentration ranges, i.e., 0 to 25 mM in urine and 0 to 25 µM in serum, using the internal standard at a fixed concentration of 1000 µM for d_6_-metformin (d_6_-METF) in urine and 20 µM in serum. SIM of *m*/*z* 383 for d_0_-METF and *m*/*z* 383 for d_6_-METF was performed as described previously [19]. The results of this experiment are summarized in Table 5. A typical GC-MS chromatogram from the analysis of metformin in a human serum sample is shown in Figure 3.

The PA of *m*/*z* 389 for d_6_-METF-PFP varied by 31% in U and by 30% in S. The retention times of the isotopologs differed in both matrices. The IE and δ values differed statistically significantly between U and S.

## 4. Discussion

The Bland–Altman approach has been proposed for testing agreement between two measurements [4]. The graphical Bland–Altman approach is frequently used in analytical chemistry to compare two analytical methods for the quantitative determination of analytes in biological samples [5]. The ROC approach is also a graphical plot that is often used to measure differences between two methods of measurement of analytes, although the main aim of this approach is testing disagreement between two approaches, especially in clinical diagnosis [33]. Given the potentially very small differences in the retention times of isotopologs in chromatography [1,2,3], we investigated, in the present study, the utility of the Bland–Altman and ROC approaches in the GC-MS analyses of selected endogenous analytes in human plasma, serum and urine samples. The focus of the study was on the main members of the L-arginine/nitric oxide pathway [11] and creatinine, which is an important clinical biochemical parameter. 

Nitrate and nitrite and the externally added ^15^N isotopologs were analyzed by GC-MS after derivatization with PFB bromide to their PFB-ONO_2_ and PFB-NO_2_ derivatives, respectively (Figure 1). PFB-ONO_2_ and PFB-NO_2_ are separated completely by GC as well as by MS. Being a nitric acid ester, PFB-ONO_2_ eluted in front of PFB-NO_2_, which is a nitro derivative [13]. Virtually, both the Bland–Altman and the ROC approach are not able to discriminate ^14^N/^15^N isotopologs of PFB-ONO_2_ and PFB-NO_2_, respectively. Yet, small differences were detected in plasma, serum and urine samples, independent of the extraction solvent that contained the derivatives, i.e., toluene and ethyl acetate. In contrast, both the Bland–Altman approach and the ROC approach clearly discriminated the respective H/D isotopologs of DMA (PFBz-DMA), ADMA (Me-PFP), Arg (Me-PFP) and creatinine (PFB-creatinine) (Figure 1), yet with some differences for ADMA between plasma and urine. The strong correlation found between the Bland–Altman and the ROC approaches suggests that both methods are virtually equally suitable to investigate isotope effects in GC-MS.

The two methyl groups of DMA in its PFBz derivative, the methyl group of creatinine in its PFB derivative and the methyl ester groups of Arg and ADMA in the methyl ester PFP derivatives are most likely responsible for the considerably stronger H/D isotope effects compared to the ^14^N/^15^N isotope effects observed in PFB-ONO_2_ and PFB-NO_2_ derivatives. The greater differences in physical properties between H and D (a 100% increase in mass) compared to the differences between ^14^N and ^15^N (a 7% increase in mass) are a likely explanation for the stronger H/D isotope effects. 

The charge radius of D is 2.5 times higher compared to the charge radius of H (https://physics.nist.gov/cuu/Constants/index.html, assessed on 10 December 2023). The gravest factor that causes the stronger H/D isotope effects is likely to be a stronger interaction of the methyl groups with the lipophilic stationary phase of the GC column (50% methylpolysiloxane, 50% phenylpolysiloxane) in the present study. H/D effects were observed for non-derivatized methylxanthine isotopologs in GC-MS on a 14% cyanopropylphenyl methylpolysiloxane fused silica column [34]. In that study, H/D isotopic effects were found to depend not only on the number of D atoms but also on the position of the CD_3_ groups in the molecules (IE range, 1.00147 to 1.00668) [34]. The rate of a reaction involving a C–H bond is typically 6–10 times faster than the corresponding C–D bond [35].

In the case of PFB-ONO_2_ and PFB-NO_2_, the central N atoms seem to be strongly sterically hindered from interacting with the stationary phase. The differences seen between PFB-ONO_2_ and PFB-NO_2_ suggest that the N atom in PFB-ONO_2_ is somewhat more accessible to interaction with the stationary phase than the N atom in PFB-NO_2_, which is closer to the PFB group (Figure 1). This observation demands deeper investigations with nitro and nitric acid derivatives of alkyl/aryl residues.

In the cases of ADMA, nitrate, nitrite and creatinine, which were analyzed both in plasma/serum and in urine, there were some differences in the Bland–Altman δ (%) and ROC-AUC values in plasma or serum compared to urine. ADMA: 0.1409 vs. 0.1298 (1.1-fold); nitrate: 0.06909 vs. 0.04029 (1.7-fold); nitrite: 0.00841 vs. 0.02891 (0.3-fold). These observations may be interpreted as a type of “matrix effect”. Matrix effects occur not only in LC-MS/MS but also in GC-MS and GC-MS/MS [22,23,24,25,26,27,28,29,30,31,32]. Several methods have been proposed and used in LC-MS/MS, such as the use of standard line slopes as a measure of relative matrix effects [22]. The results of the present study, including those of the pilot experiment, indicate that the differences in the retention times of isotopologs δ are better suited to quantify matrix effects than the ratio of the retention times IE of d_0_-DMA-PFBz and d_6_-DMA-PFBz in human urine. IE is a little variable measure but is less sensitive than the more variable measure δ. The Bland–Altman approach seems to be better suited for quantitating isotope effects than the ROC approach. 

## 5. Conclusions

In GC-MS, the Bland–Altman and ROC approaches seem to be suitable for studying H/D and ^14^N/^15^N isotope effects in the PFB, PFBz and PFP derivatives of endogenous analytes of the L-arginine/nitric oxide pathway and the universal biomarker creatinine. Isotope effects in GC-MS are likely to be caused by differences in the interaction strengths of H/D and ^14^N/^15^N isotopes in the derivatives with the hydrophobic stationary phase of the GC column. D atoms in the derivatives seem to attenuate the interaction of the skeleton of the molecules with the lipophilic GC stationary phase. Differences in the retention times of isotopologs, i.e., the parameter δ, appear to be a better-suited experimental measure for quantitating matrix effects in GC-MS.

## Data Availability

The data presented in this study are available in article.

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
