# Peer review of "Application of the Bland–Altman and Receiver Operating Characteristic (ROC) Approaches to Study Isotope Effects in Gas Chromatography–Mass Spectrometry Analysis of Human Plasma, Serum and Urine Samples"

_molecules, 2024, doi:10.3390/molecules29020365_

Round 1

Reviewer 1 Report

Comments and Suggestions for Authors

Reviewer report on manuscript Molecules-2795424

The submitted experimental work aimed towards the application of the Bland-Altman and Receiver Operating Characteristic (ROC) Approaches to Study Isotope Effects in GC-MS Analysis of Human Plasma, Serum and Urine Samples.

I read the manuscript carefully. In general terms, the manuscript is well-written, and the method is well-validated. The research has sufficient novelty to justify publication in the current Journal. Finally, I recommend its acceptance.

Other comments

1)      Line 107: The dot symbol should be deleted.

2)      The schemes and figures’ captions should be reduced somehow.

3)      Table 1: I would recommend deleting the “dwell time” column by inserting footnote to improve the readability of the table.

4)      Table 5 could ne uploaded as supplementary material (as landscape) to improve its visual readability.

5)      Section 3.2.1: what is the role of sodium carbonate in the derivatization reaction?

6)      The introduction should be strengthened somehow by presenting the novelty of the proposed approach.

Author Response

Reviewer report on manuscript Molecules-2795424
Reviewer #1
The submitted experimental work aimed towards the application of the Bland-Altman and Receiver Operating Characteristic (ROC) Approaches to Study Isotope Effects in GC-MS Analysis of Human Plasma, Serum and Urine Samples.
I read the manuscript carefully. In general terms, the manuscript is well-written, and the method is well-validated. The research has sufficient novelty to justify publication in the current Journal. Finally, I recommend its acceptance.

Response

Other comments
1) Line 107: The dot symbol should be deleted.

Response: deleted

2) The schemes and figures’ captions should be reduced somehow.
Response
I believe that the captions are appropriate

3) Table 1: I would recommend deleting the “dwell time” column by inserting footnote to improve the readability of the table.
Response
deleted

4) Table 5 could ne uploaded as supplementary material (as landscape) to improve its visual readability.
Response
I think, the journal will manage well Table 5.

5) Section 3.2.1: what is the role of sodium carbonate in the derivatization reaction?
Response
Thank you. The role of carbonate is explained.

6) The introduction should be strengthened somehow by presenting the novelty of the proposed approach.
Response
It was strengthened a little.

Reviewer 2 Report

Comments and Suggestions for Authors

The article molecules-2795424 is devoted to the application of the Bland-Altman method and ROC to study isotope effects in gas chromatography. I have a number of small comments about the description of the methodology and design, but they are not critical.

1) In the methodology of a gas chromatographic experiment, it is necessary to indicate the temperature conditions of the study. Were all experiments carried out under the same conditions?

2) There are typos in the text, for example in the title of paragraph 2.2.

3) Is it possible to provide examples of chromatograms and mass spectra in additional materials?

Author Response

Reviewer #2
The article molecules-2795424 is devoted to the application of the Bland-Altman method and ROC to study isotope effects in gas chromatography. I have a number of small comments about the description of the methodology and design, but they are not critical.

Response

1) In the methodology of a gas chromatographic experiment, it is necessary to indicate the temperature conditions of the study. Were all experiments carried out under the same conditions?
Response
The oven temperature program was reported for metformin. This program was used for amino acids.

2) There are typos in the text, for example in the title of paragraph 2.2.
Response
Corrected

3) Is it possible to provide examples of chromatograms and mass spectra in additional materials?
Response
Thank you. The new Figure 3 is provided as an example.